# Oceanic drivers of UK summer droughts

Amulya Chevuturi [1] ✉, Marilena Oltmanns [2], Maliko Tanguy [1,3], Ben Harvey [4],
Cecilia Svensson [1] & Jamie Hannaford [1,5]

UK droughts are projected to become more frequent under climate change, reinforcing the need to understand their underlying causes. Our study examines oceanic drivers of UK summer droughts and the associated teleconnection pathways. Specifically, we evaluate statistical links between standardized precipitation and streamflow indices for the UK and two North Atlantic Sea surface temperature (SST) patterns which have previously been linked to the influx of freshwater into the subpolar region. Our findings reveal that the North Atlantic SST influences UK hydrology up to 1.5 years in advance by altering the position of the North Atlantic Current, which is coupled to the location of the North Atlantic summer jet stream. The long lead time of this teleconnection pathway can inform UK drought forecasting across seasonal to interannual timescales and ultimately contribute to the advancement of sustainable water resource management in the face of increasing drought risks in the UK.

Extreme droughts have significant impacts on many sectors, including public water supply, agriculture, and energy production[1,2], as well as on terrestrial and freshwater ecosystems. UK future climate projections indicate a rise in droughts[3,4], especially during the summer[5], albeit with associated uncertainties[6,7]. The potential for an enhanced risk of drought in the future has profound implications for sustainable water resource management[8].

UK droughts have long been associated with North Atlantic oceanic and atmospheric variability. Case studies of extreme droughts across Europe show that the North Atlantic Oscillation (NAO) and Eastern Atlantic/Western Russia patterns influence the droughts' spatio-temporal evolution[9,10]. These atmospheric patterns dominate in winter and have the greatest impact during the same season by reducing rainfall. This deficiency in rainfall can extend into the subsequent summer, contributing to low streamflow patterns that can eventually culminate into multi-year extreme drought events. Studies have further analysed the long lead-time predictive capacity of NAO signals on water resource droughts in the UK[11,12]. Case studies have also highlighted a consistent tripolar sea surface temperature (SST) pattern in the North Atlantic during the summer preceding extreme UK droughts[13]. This tripole pattern exhibits a meridional temperature gradient reminiscent of the negative phase of the Atlantic multidecadal variability mode[14]. Moreover, the summer tripole pattern was found to be linked to NAO-like circulation patterns in the subsequent winter[15–17]. These changes over the North Atlantic ultimately contribute to lower-than-normal rainfall in the UK during subsequent summers through a poleward shift of the eddy-driven jet[13,18].

The atmospheric bridge link between the North Atlantic tripole pattern and UK rainfall described above has a lag of 1 year[19]. However, recent research[20] suggests that influx of freshwater in the subpolar region of the North Atlantic Ocean during summer has been linked to warm and dry weather conditions over Europe during the summer 2 years later, which indicates a teleconnection pathway over an even longer period. Despite the great societal value in such long lead times, allowing for longer-term planning and preparation, the impact of these events on hydrological droughts, especially over the UK, is still unknown.

The current challenge with seasonal hydrological forecasts over UK[21] lies in their limited skill at long lead times, with exceptions over southeast UK[22]. Even then, skill primarily emerges from initial hydrological conditions (especially storage in groundwater), whereas skill of the driving operational meteorological forecasts is relatively limited. Although meteorological forecasts skill for winter has advanced[23] improving winter hydrological forecasts[24], skill in other seasons, especially summer, remains limited[25]. Compared with atmospheric processes, oceanic processes have more power at lower (interannual and decadal) frequencies, presenting a possible avenue to use teleconnections from the North Atlantic Ocean for longer-range drought predictions. Development of early warning systems from such teleconnections offers a great opportunity for sustainable and reliable water resource management across various sectors such as agriculture, hydro-electric power, and urban water supply planning[26]. Improved understanding of the ocean-atmosphere processes driving extreme droughts can underpin improved water resources management more generally. For example, long-term water resource planning in the UK (as elsewhere) relies on the use of stochastic simulation to generate droughts more extreme than in observed

[1]UK Centre for Ecology & Hydrology, Wallingford, UK. [2]National Oceanography Centre, Southampton, UK. [3]European Centre for Medium-Range Weather Forecasts, Reading, UK. [4]National Centre for Atmospheric Science, Department of Meteorology, University of Reading, Reading, UK. [5]Irish Climate Analysis and Research UnitS (ICARUS), Maynooth University, Maynooth, Ireland. ✉e-mail: amuche@ceh.ac.uk

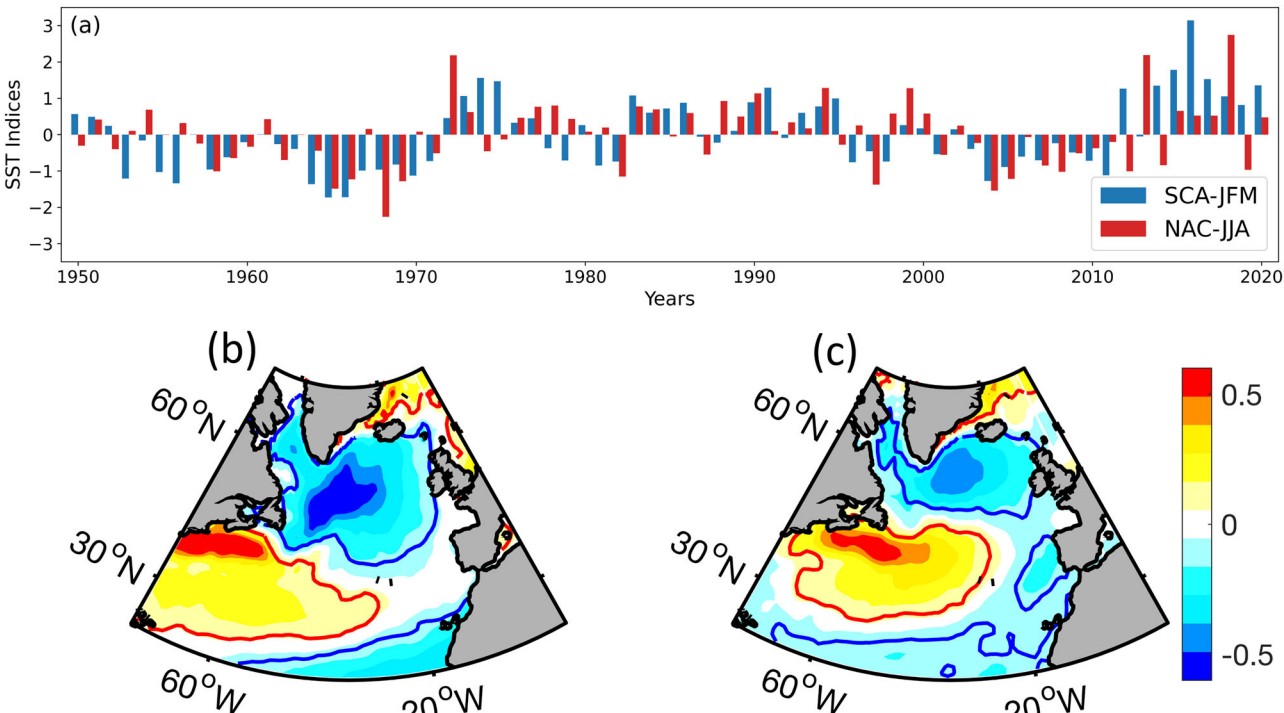

**Fig. 1 | Spatial and temporal variability of North Atlantic Ocean patterns. a** Yearly time series for the SCA-JFM index (blue) and NAC-JJA index (red) as bars. SST seasonal anomaly patterns calculated as regressions (shaded; °C) against the **b** SCA-JFM index and **c** NAC-JJA index over 1950—2022.

records[27] and these approaches currently rely on atmospheric/oceanic predictors such as the NAO[28]. New knowledge of driving mechanisms could thereby underpin improved simulation of extreme events to provide stress tests for long-term water resources planning.

While previous studies have explored teleconnections from the North Atlantic SST to explain meteorological droughts[13], the connection to hydrological (streamflow) droughts remains largely unexplored. Climate drivers such as precipitation directly influence a region's hydrology, while temperature affects it via snow accumulation or melting and evaporative demand[29,30], which is further influenced by wind speed and cloudiness[31]. However, despite the strong link between a region's meteorology and hydrology[32], not all meteorological droughts propagate into hydrological droughts[33]. Therefore, studying the influence of a teleconnection only on rainfall is insufficient[34] to fully understand its impacts on water resources. Rivers integrate processes such as precipitation and evapotranspiration over a catchment[35,36], influenced by catchment characteristics[37,38], resulting in hydrological responses that differ from those of their meteorological drivers[39–41]. For example, in regions with significant groundwater influence, such as the southeast UK, groundwater contributions to river flows can buffer the effects of reduced precipitation, preventing the development of hydrological droughts[39], on the other hand, percolation of rainfall to groundwater stores prevents its immediate contribution to river flow[35,36]. As hydrological droughts are important indicators of water availability and the impacts of droughts[42,43], they are directly relevant for operational water resource management[12,44]. Thus, it is imperative to study the impact of teleconnections directly on water resource deficits, rather than simply assuming they are a direct consequence of meteorological droughts[33,38,40], and relying solely on temperature or precipitation as proxies for hydrological droughts.

Moreover, there exists a significant underestimation of the predictable climate signal within dynamical models, notably pronounced in the Arctic–North Atlantic region, attributed to the signal-to-noise paradox[45]. Potential reasons for this underestimation include errors in ocean–atmosphere coupling, teleconnections and parameterised processes at current model resolutions[46]. Current models have large biases in the

freshwater and hydrographic variability of the subpolar North Atlantic (e.g.[47–50]). Further, previous studies have focused on individual components of the teleconnection from North Atlantic freshwater events to UK hydrology (e.g.[13,17,18]), rather than analysing the entire chain of interactions necessary for its integration into modelling frameworks. By tracing the full pathway of this teleconnection to hydrological droughts in the UK, we not only attribute causation to statistical links but also instill confidence in prediction systems built upon these relationships. Forecasts from such systems have direct implications for water resource planning, as accurate early predictions are essential for water managers to prepare for and mitigate the impacts of droughts in advance. Therefore, this study aims to improve our understanding of the predictive capacity for long-lead-time droughts over the UK by significantly increasing our understanding of the driving teleconnections through:

i. Establishing statistical links between the variability in the North Atlantic Ocean and UK hydrometeorology.
ii. Assessing how changes in North Atlantic salinity and temperature drive the development of extreme UK summer droughts in the following year.

## Influence of North Atlantic SSTs on UK hydrometeorology

Motivated by the observed link of northern European droughts with North Atlantic subpolar cold anomalies (SCA) and subsequent northward North Atlantic Current (NAC) shifts[18], we construct two SST indices, SCA index for winter (SCA-JFM) and NAC index for summer (NAC-JJA) (Fig. 1a) describing the two SST patterns that correspond to the cold anomalies and the NAC shifts (Fig. 1b, c; see 'Methods' for details).

We perform correlation analysis between the area-averaged standardised precipitation index (SPI) for 3-month accumulations (SPI3) in two distinct regions of UK rainfall, northwest (NW) and southeast (SE) (Fig. 2a), and the two SST indices (Fig. 1a). Significant correlations (Fig. 2b) are seen between the SCA-JFM and SPI3, at a lag of 0 and then at 1.5 years. Similarly, the NAC-JJA displays significant correlations with SPI3 at lag 0 and then at lag of 1 year (mostly NW UK). Thus, both the summer and winter SST

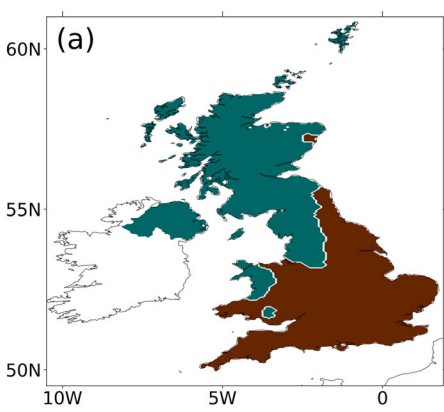

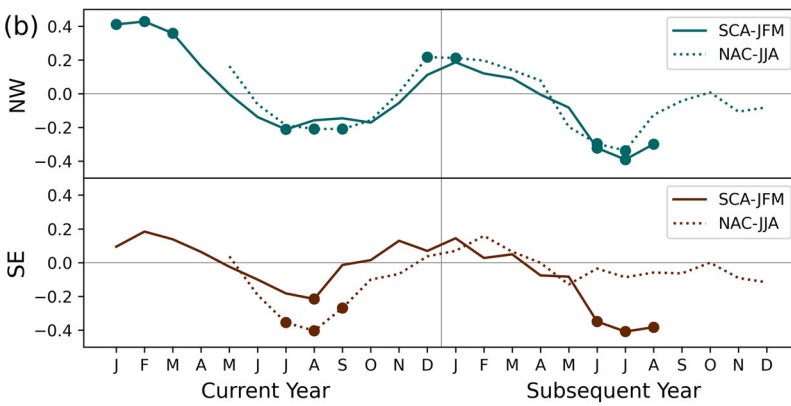

**Fig. 2 | Lagged correlations between North Atlantic Ocean indices and regional UK precipitation. a** UK divided into two regions of SPI3 northwest (green) and southeast (brown) using k-means clustering. **b** Correlation coefficients between two SST indices: SCA-JFM (solid line) and NAC-JJA (dotted line) against regionally averaged SPI3 over the two regions northwest (green) and southeast (brown) for each month in the current year and the subsequent year. Round markers signify correlations that are significant at the 0.05 level with lags shown up to 18 months.

indices are significantly correlated with the UK summer rainfall at different lags. These connections are further linked by the significant correlation between the SCA-JFM and the North Atlantic jet position index for the subsequent year's JJA (correlation coefficient = 0.34), and between NAC-JJA and the jet index of the same year's JJA (correlation coefficient = 0.31). North Atlantic jet stream position is known to strongly influence UK rainfall[51], and its correlation to SST indices suggests a potential link in the teleconnection chain between the North Atlantic SSTs and UK climate. Area-averaging the SPI and jet stream over large regions may weaken correlation signals due to yearly spatial variations in rainfall patterns and jet stream positioning. To address this, we extend our analysis to examine local correlations (Fig. 3) and investigate specific case study years (Section 3).

Although each SST index is strongly auto-correlated with itself at the lag of 1 year (SCA-JFM = 0.56 and NAC-JJA = 0.41), there is no notable autocorrelation in the regional SPI3 time series, as these SST indices do not account for its full variance, ruling out the possibility of spurious correlations. Moreover[20], demonstrated that the link between North Atlantic SSTs and European summer weather is significant on all timescales from years to decades.

The winter SCA-JFM index also exhibits a significant correlation with the summer NAC-JJA index of the next year (correlation coefficient = 0.45). The link between the SST patterns has been explained by ocean-atmosphere feedbacks in previous research[20]. The SCA-JFM is associated with a positive NAO signal and hence, stronger westerly winds. The specific location of the wind anomaly, associated with the subpolar cold anomaly signal, leads to converging Ekman transports (i.e. the wind-driven component of the ocean current) in the intergyre region[52]. The convergence of the surface oceanic flow sets up geostrophic pressure gradients that are maintained through the following summers despite the winds abating. The geostrophic flow associated with these pressure gradients reflects a northward shift of the NAC, leading to the characteristic warm anomaly in the intergyre region. This dynamical relationship is supported by the positive correlation between the SCA-JFM and the NAO index for JFM concurrently (within the same year, correlation coefficient = 0.32), as well as the positive correlations of both SST indices with the NAO JFM of the subsequent year (correlation coefficients with SCA-JFM = 0.31 and NAC-JJA = 0.33). The winter NAO has previously been statistically linked to British summer river flows[53], for a study period dominated by the cold phase of the Atlantic multidecadal oscillation.

While the exact timing of the onset of the northward NAC shift pattern can vary depending on the duration of the cold and fresh anomaly in the subpolar region, the average lag between the subpolar cold anomaly in winter of the previous year and its impact on UK

hydrology in the summer of the following year is ~1.5 years. Thus, these feedbacks, operating at a relatively extended time scale, and their long lead time association with UK hydrology, provide an opportunity to predict UK droughts well in advance.

Inspecting the spatial distribution of the correlation between the SST patterns and the SPI, significant links are identified. Significant correlations exist across the entirety of the UK between the SCA-JFM index and the SPI during summer of the subsequent year (Fig. 3a). For the NAC-JJA index, significant correlations are observed with SPI over most of the UK except some parts in the northwest (Fig. 3b).

The regions showing significant correlations between the SST indices and the standardised streamflow Index at a 3-month accumulation (SSI3) for next year's summer (Fig. 3c, d) are largely consistent with those identified for the SPI3 (Fig. 3a, b). Hence, this establishes a statistical link between the North Atlantic SCA and streamflow over most of the UK. However, there are a couple of differences between the SSI3 and SPI3 patterns. First, SSI3 in the southeastern regions does not exhibit significant correlations with either index, in contrast to the correlation patterns with SPI3. This difference can be attributed to the nature of catchments in the southeastern UK, where the hydrological behaviour of catchments is primarily groundwater driven and consequently, these catchments are slow responding, and not predominantly influenced by precipitation at short time scales[37,54]. Second, the area of poor correlations in northwest Britain for SPI3 (Fig. 3b) has expanded to encompass most of western Britain for SSI3 (Fig. 3d). The areas showing a good relationship between NAC-JJA and SSI3 only occur well into to the predominantly leeward, well sheltered east side of the topographical divide. These catchments lack headwaters near the actual topographical divide and are therefore less susceptible to drift of orographically enhanced precipitation across the divide. The differences between the impacts of NAC-JJA on SPI3 (Fig. 3b) vs SSI3 (Fig. 3d) could also be due to high evaporative demand, especially in summer, which reduces the proportion of rainfall contributing to river flow. Increasing evaporative demand in spring, as found by ref. 55, could lead to drier soils, causing summer precipitation to first replenish soil moisture, with some of it lost to evaporation rather than reaching rivers.

The ensemble streamflow prediction (ESP) method is employed operationally within the UK Hydrological Outlook[21] to generate seasonal hydrological forecasts for the UK. While the ESP exhibits some skill in catchments in the southeast of the UK for lead times of up to 12 months, this skill decays over time[22]. Additionally, its overall performance is poor in regions outside the southeast UK at similar lead-times. The strong correlations observed between the North Atlantic SST patterns and streamflow across most of the UK, excluding the southeast, may be used to compensate for the limitations in the current operational forecasting system.

**Fig. 3 | Lagged correlations between North Atlantic Ocean indices and UK summer hydrology.** Correlation coefficient (shaded) between SST indices **a** SCA-JFM (year-0) and **b** NAC-JJA (year-0) against SPI3 for JJA in the subsequent year at each grid point. **c, d** same as (**a, b**) but for correlations against SSI3 for JJA in the subsequent year at each catchment. Grey regions demonstrate regions showing insignificant correlations at the 0.05 level.

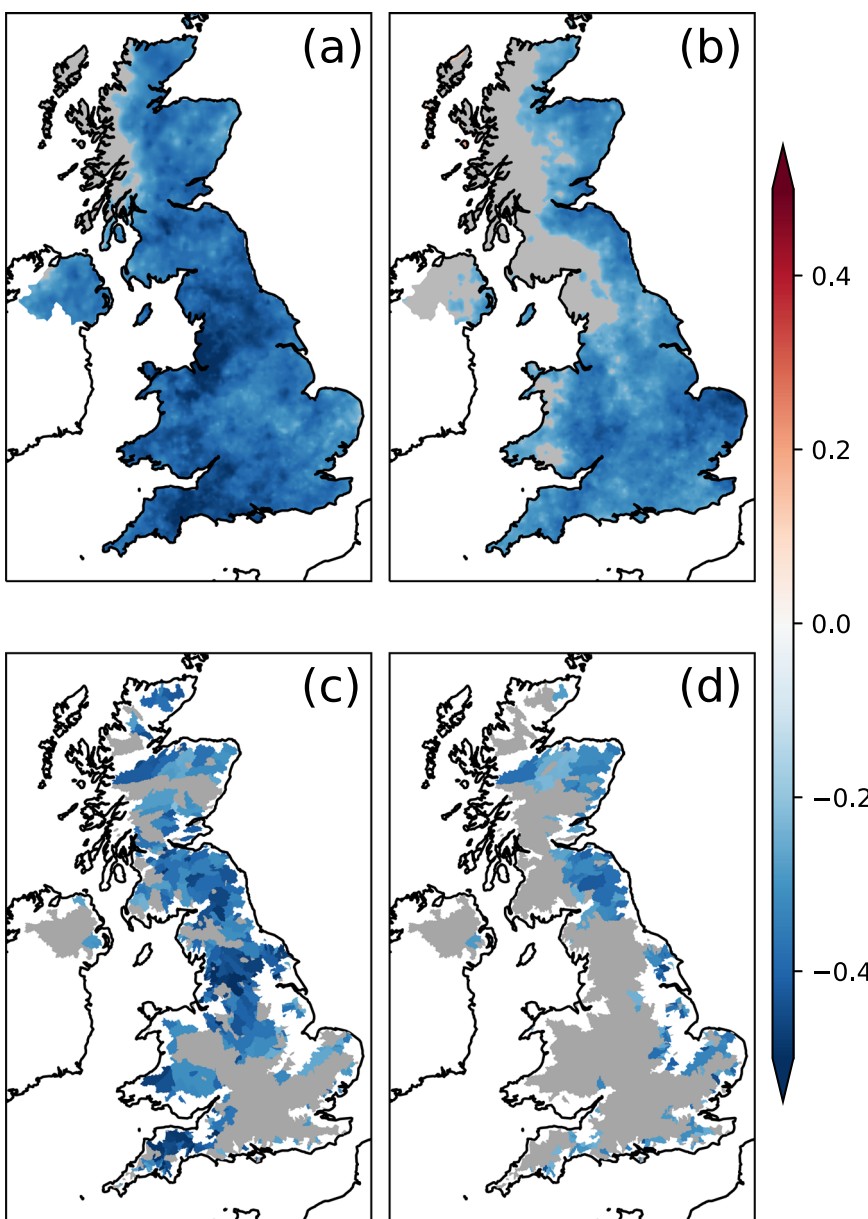

Consequently, these correlations present an opportunity to develop a reliable nationwide early warning system that extends beyond the 1-year lead time.

## Case studies of UK droughts

Having established strong statistical links between the North Atlantic SST and UK droughts, we aim to trace back the teleconnection pathways leading to extreme summer UK drought events. We identified the 4 years with the highest positive values of the NAC-JJA index (Fig. 1a), i.e. with the most northward position of the NAC. These years experienced four of the most extreme summer droughts: 1976, 1995, 2018 and 2022. The 1976 drought is widely regarded as the most severe drought in recent UK history[56]. The 1995 drought was considered as the second most severe drought[57], particularly impacting the southeast region, but also caused serious nationwide effects, including major water supply issues in northern England[58,59]. More recently, the droughts in years 2018 and 2022 were some of the severest summer droughts to affect the UK[2,60,61].

For these four drought years, the precipitation (SPI3; Fig. 4a–d) and streamflow (SSI3; Fig. 4e–h) are below normal and are associated with anomalously high mean sea level pressure (Fig. 4i–l) and with weakening of the prevailing westerlies (negative anomalies of u-component of wind over the UK; Fig. 4m–p). The northward position of the positive anomalies in the u-wind anomalies at 850hPa compared to its JJA-mean climatology, implies poleward shifted North Atlantic jet stream during these 4 years. Such poleward shifts in the North Atlantic jet have long been correlated with dry conditions over the UK[51].

The anomalously northward positioning of the NAC (Fig. 5a–d) leads to the northward jet position (Fig. 4m–p) and is in turn influenced by the westerly winds associated with the positive phase of the NAO during the preceding winter. The positive NAO phase is associated with colder (warmer) temperature anomalies in the subpolar (subtropical) North Atlantic (Fig. 5e–h; also in refs. 15,62).

In the winter, 1.5 years before each of the summer droughts, cold anomalies were observed over both the North Atlantic subpolar gyre and the tropical North Atlantic, along with warm anomalies over the mid-North Atlantic (Fig. 5i–l). This SST pattern bears a striking resemblance to the tripole pattern[15] and has been linked to European droughts in 1976 and 2018[13]. This tripole SST pattern (Fig. 5i–l) is associated with the positive SCA-JFM index (Fig. 1b) and has further been linked to freshwater anomalies in the subpolar North Atlantic[20]. When we consider

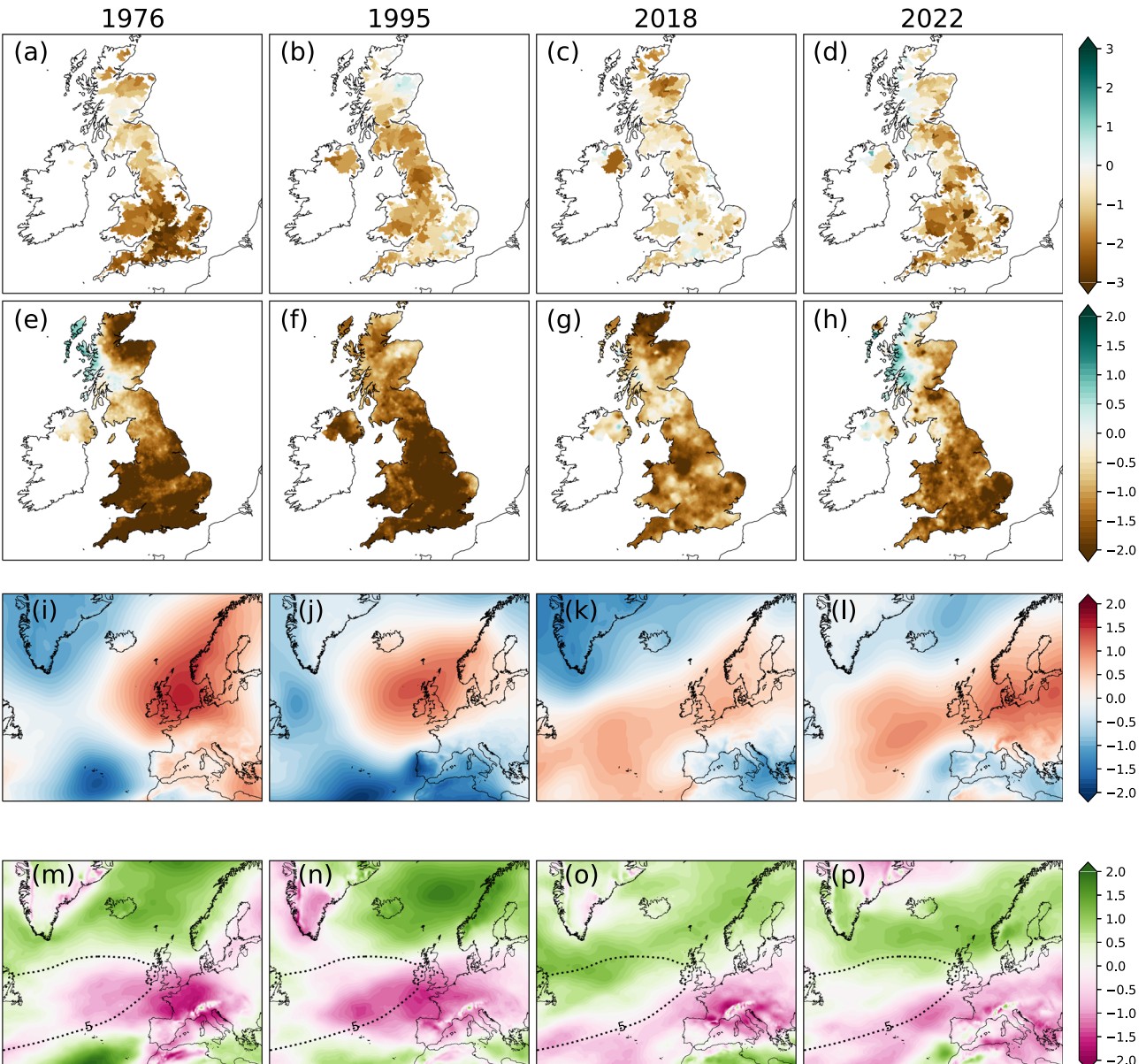

**Fig. 4 | Hydrometeorological conditions during extreme UK summer droughts.** Four extreme summer drought years' case studies for (**a, e, i, m**) 1976, (**b, f, j, n**) 1995, (**c, g, k, o**) 2018 and (**d, h, l, p**) 2022, showcasing JJA summer (**a–d**) SSI3 (shaded), (**e–h**) SPI3 (shaded), (**i–l**) mean sea level pressure anomalies (shaded, Pa), and (**m–p**) 850hPa u-component of wind anomalies (shaded, m/s) with the 8500hPa u-component of wind climatology from 1950 to2020 (contour, 5 m/s).

the latest three drought events, the sea surface salinity (SSS) shows negative anomalies over most of the subpolar North Atlantic region (Fig. 5m–o), indicating lower than normal salinity over the region. Observed gridded SSS data is not available as far back as 1974, however, there was a freshening event during that period which multiple studies have called North Atlantic 'Great Salinity' which extended from 1968 to 1982[63].

While the four drought cases share similarities, notable regional variations are evident within each year, particularly in the northward extent of the North Atlantic jet stream and NAC. We attribute the differences in the exact location of the NAC to differences in the size, extent and location of the preceding freshwater and cold anomaly patterns, which influence the subsequent feedback chain[20].

## Discussion and conclusions
Our analysis reveals robust statistical links between specific North Atlantic SST patterns—influenced by freshwater influx into the subpolar North

Atlantic—and UK hydrometeorology, at a lag of 1.5 years. These SST patterns demonstrate significant correlations with precipitation across the entire UK and streamflows in most regions, except the groundwater driven catchments in the southeast.

To our knowledge, this is the first study to go beyond the statistical relationships, and trace the complete teleconnection pathway from North Atlantic freshwater changes to UK hydrology, using four historic extreme UK summer droughts as case studies. We propose the following teleconnection pathway, which is illustrated in Fig. 6 and supported by earlier studies that demonstrate individual links. Firstly, we attribute the four extreme UK summer droughts to the anomalous northward shift of the North Atlantic jet stream, which causes the North Atlantic storm track to move further north, resulting in reduced rainfall over the UK (as also seen in refs. 13,51). The northward shift in the jet (or the large-scale atmospheric circulation associated with the jet stream shift) is in turn closely linked to northward shift of the NAC[64]. This is associated with more northward SST front between the warm NAC and the colder subpolar region[20], resembling

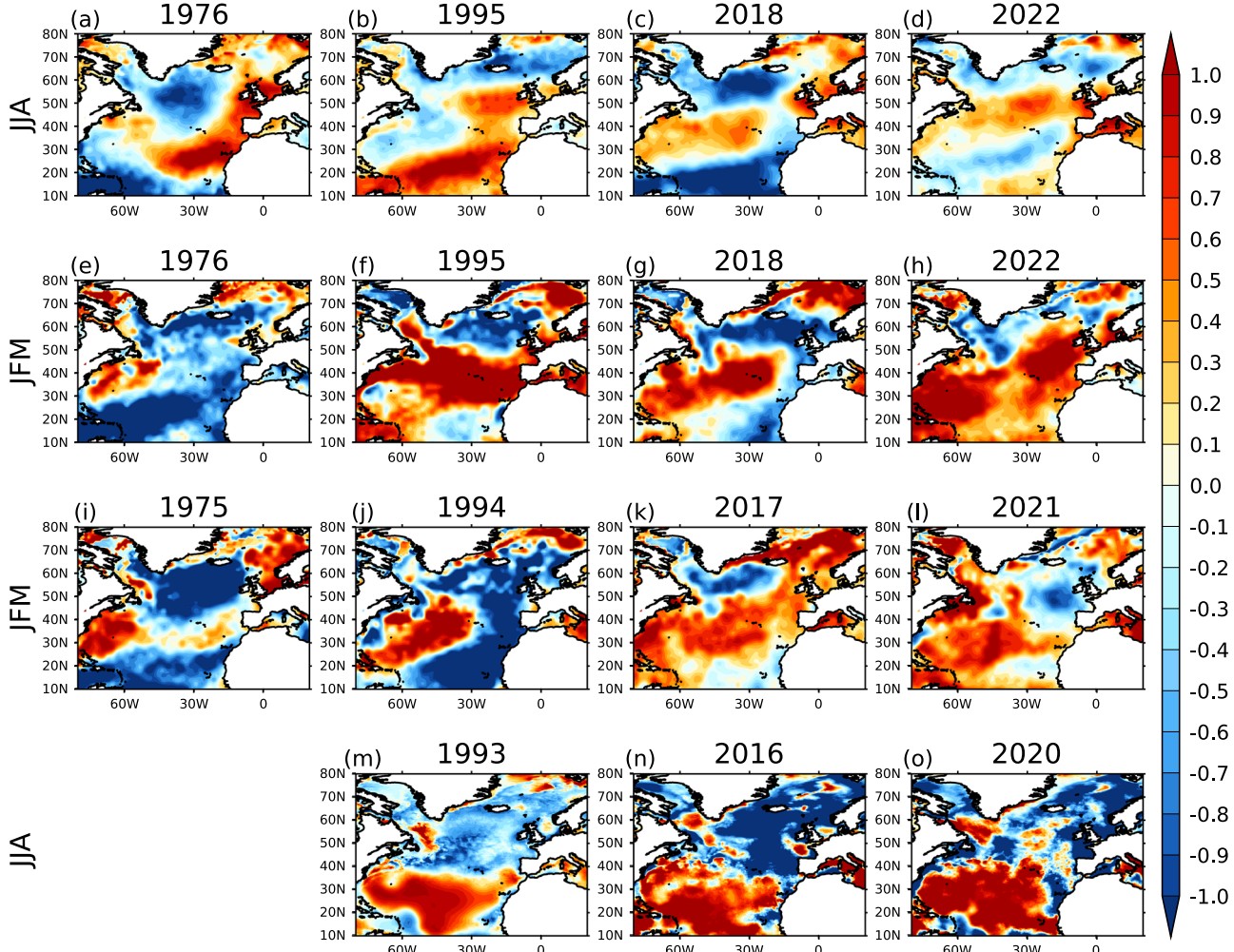

**Fig. 5 | North Atlantic Ocean conditions linked to extreme UK summer droughts.** Standardised anomalies for (**a–d**) JJA SST for the same year as the droughts 1976, 1995, 2018 and 2022 (lag 0 year), (**e–h**) JFM SST for the same year as the droughts 1976, 1995, 2018 and 2022 (lag −0.5 year), (**i–l**) JFM SST one year prior to the droughts: 1975, 1994, 2017 and 2021 (lag −1.5 year), and (**m–o**) JJA SSS two years prior to the droughts: 1993, 2016 and 2020 (lag −2 year). Standardised anomalies for SSTs are calculated from detrended SSTs.

the positive phase of the NAC-JJA index. The northward shifts in the North Atlantic jet and NAC have been previously linked to the winter NAO circulation[10,65]. The winter NAO is preceded by the tripole SST pattern[18,66], which resembles the positive phase of the SCA-JFM index (Fig. 1b), characterised by cold anomalies over both the subpolar gyre and the tropical North Atlantic, with warm anomalies in the intervening region[13]. The westerly winds associated with positive NAO circulation pattern along with the SCA of the North Atlantic tripole pattern can explain the northward shift of the NAC[20,52,67].

Further, the development of SCA-JFM (or the North Atlantic tripole) SST pattern in winter is linked to an enhanced freshwater influx into the subpolar regions of North Atlantic[20,68]. Specifically, increased surface freshening of the subpolar North Atlantic during summer increases the stratification of the ocean and prevents mixing. In subsequent autumn and winter, enhanced surface cooling is required to make the fresher surface water dense enough to sink. Thus, cold surface anomalies can emerge, leading to atmospheric feedbacks that intensify the signal and transform it into the characteristic large-scale ocean response that resembles the tripolar SST pattern[68]. With further feedback from NAO westerlies, the tripolar SST pattern gradually transitions into the northward NAC shift, leading to the conditions that promote strong UK summer droughts.

In summary, the proposed chain of events is well supported by the identified, significant statistical relationships and well-grounded in theory. Specifically, the link between freshwater and the SST is a result of

mass conservation. The influence of the SST on the atmospheric circulation can be explained by baroclinic instability[69]. The feedback of the atmosphere on the ocean circulation is the result of Ekman transports and geostrophic balance, and the link between the large-scale atmospheric circulation in summer and UK droughts is supported by quasi-geostrophic theory.

Our findings offer new insights and opportunities for predictions since the existing operational seasonal streamflow forecasts for the UK exhibits low skill outside the southeast[22]. Crucially, the relationships described herein provide potential hydrological forecasting options for the summer months at long lead times, that are currently one of the key shortfalls in dynamical seasonal meteorological (and hence hydrological) forecasts[25]. Although there have been efforts to improve UK summer meteorological forecasts, these improvements have primarily focused on shorter lead times of around six months[70,71]. Additionally, summer is the time of year where hydrological droughts have the greatest impact on public water supply and irrigated agriculture (due to fine balances between supply and demand) as well as acute impacts on instream ecology that arise from low flows. Thus, indicators that influence UK drought conditions during summer can be leveraged to develop highly accurate prediction systems for low-flows regime. While our focus has been on UK summer droughts, the large spatial reach of the teleconnection pathway based on the associated ocean–atmosphere circulation patterns suggests that the identified links are transferable to other regions across Europe.

**Fig. 6 | Teleconnection pathway: North Atlantic salinity changes to UK summer droughts.** (i) In the summer 2 years (YEAR−2) before the droughts melting from the Arctic causes freshwater incursion into the subpolar North Atlantic Ocean leading to stratification and enhanced surface cooling in winter; (ii) strong meridional SST gradient forms, leading to atmospheric feedbacks and the formation of the North Atlantic SST tripole pattern in the winter 1 year (YEAR-1) before the droughts; (iii) positive phase of the NAO circulation pattern is formed with strong westerly winds during the winter (YEAR 0) preceding the droughts; (iv) Northward shift of the North Atlantic Current and Jet Stream leads to UK drought in the summer (YEAR 0) with lower than normal rainfall and streamflow. In the schematic the winter season has been shown with a black snowflake symbol and the summer season with a black sun symbol.

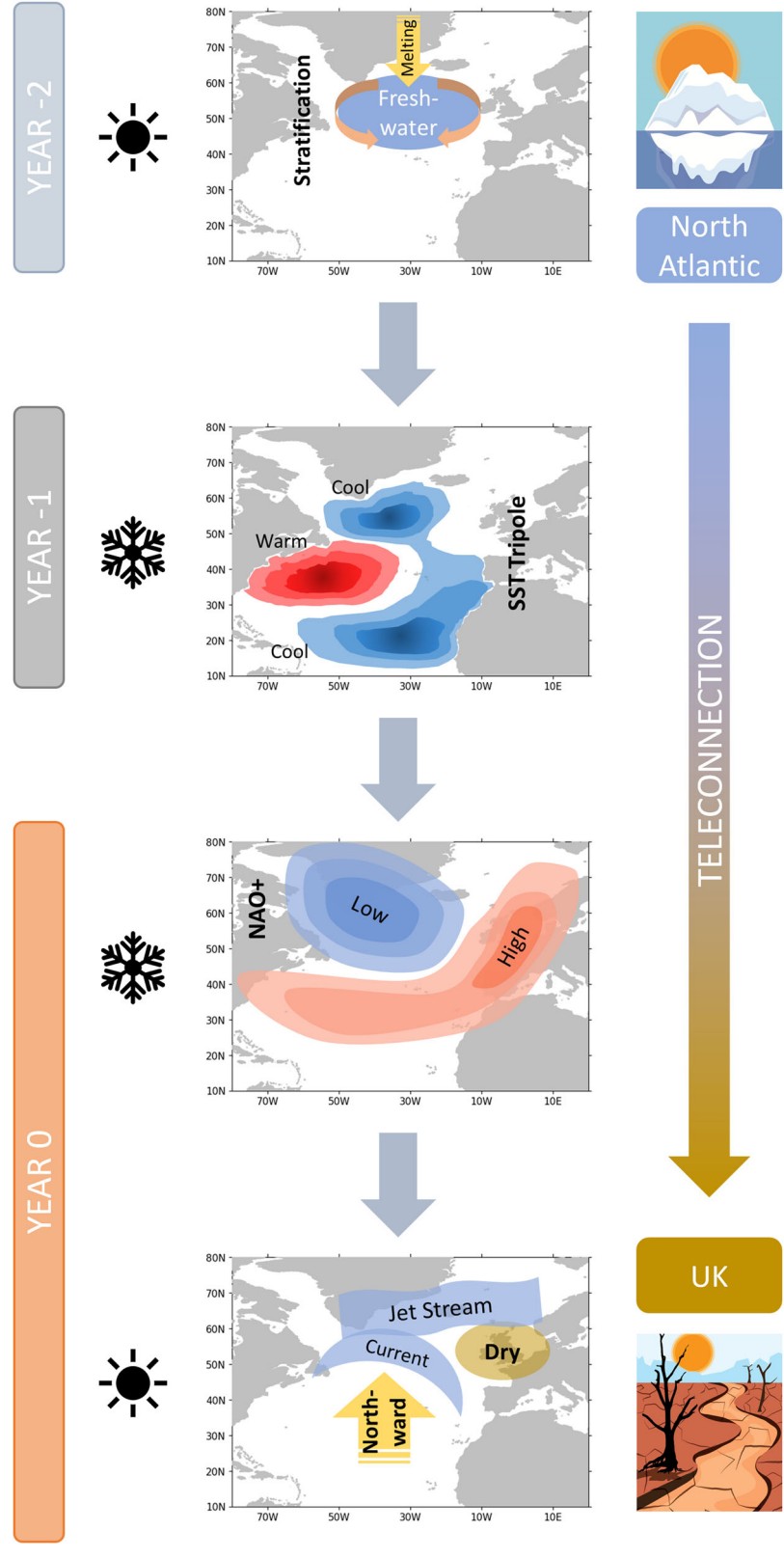

While there are only few extreme events so far that link salinity changes in the North Atlantic to UK summer hydrology, the chain of events is supported by the strong statistical relationships, by earlier studies and established theory. However, our study might be limited by the non-stationarity of correlations and trends in the observed datasets, leading to uncertainty in the statistical relationships. To identify a clear causal chain of events, SST-forced simulations can be performed with prescribed observed SST and sea ice, as demonstrated in studies like[72]. Moreover, alternative methods such as the storylines approach can be highly effective in representing uncertainty in the prolonged teleconnections being studied[73] and in exploring the boundaries of plausibility for extreme events[51]. Further, while the identified correlations in the teleconnection link explain a portion of the hydrological drought variability, it does not account for it fully. These relationships should be further refined during the development of a

predictive model by leveraging the strengths of existing forecasting method, which would enhance the overall accuracy of UK summer drought forecasts.

Forecasting systems based on SST patterns influenced by freshwater incursion can significantly enhance accuracy for low-flow regimes over UK at long lead times. Understanding the physical basis for the teleconnections is crucial before using statistical relationships to create prediction systems for UK hydrology based on North Atlantic SST/salinity changes. These predictive methods can be integrated with the existing UK hydrological forecasting system[21,22], complementing it to improve skill, particularly in northwest UK. By tracing the full teleconnection pathway to hydrological droughts, our study strengthens the causal understanding of these links, enhancing confidence in prediction systems that can be implemented operationally to support proactive water resource planning. Further, multiple predictors derived from different stages of the teleconnection pathway can be used to develop sequential drought forecasting models, which use combinations of the input predictors, which progressively improve in accuracy as the prediction period nears, as demonstrated in other parts of the world[74,75]. Improving drought forecasting systems can help mitigate the impacts of extreme droughts by enabling informed decision-making for implementing optimal water management practices[44]. This becomes especially important as global warming intensifies, leading to increased sea-ice melting, which will likely result in more freshwater influx in the North Atlantic, potentially exacerbating UK summer droughts.

## Methods
### Datasets
To characterise the hydrometeorological conditions in the UK, we use two key indices: the SPI[76] for representing precipitation patterns, and the SSI[77] to depict river flow behaviour. For this work, we used the historic gridded SPI for UK available at 5 km horizontal resolution[78,79], and observed catchment-scale SSI from 864 UK catchments[79]. To concentrate on the more significant drought events with pronounced impacts, we chose the three-month accumulations for both the indices: SPI3 and SSI3.

For SSTs we use the Met Office Hadley Centre HadISST monthly SSTs version 1.1, available at 1° horizontal resolution[80]. For sea surface salinity (SSS), we use multi observation global ocean sea surface salinity at 0.125° horizontal resolution[81]. For the atmospheric variables, u-component of wind at 850 hPa and mean sea level pressure, we use European Centre for Medium-Range Weather Forecasts fifth generation reanalysis (ERA5), available at ~31 km horizontal resolution[82]. To identify the North Atlantic jet latitude (i.e. position) during winter (DJF) and summer (JJA) months we use the corresponding jet stream indices[51], calculated using ERA5 data. The NAO winter (JFM) index used here is the station-based NAO index[83]. Our study focuses on the common period for all data variables' availability, i.e. 1950–2022.

### Derivation of drought index
The UK exhibits a strong rainfall gradient that intensifies from southeast to northwest, caused by the prevailing winds and the topographical divide. To identify homogenous rainfall regions, we apply k-means clustering method to monthly SPI3 values. Our analysis reveals two distinct clusters: the northwest and the southeast (Fig. 2a), similar to the findings by[84]. To understand the statistical relationships between SSTs patterns and UK hydrometeorology (SPI3 and SSI3) and other oceanic and circulation indices, we employ the Pearson correlation method over the common period of 1950—2020. To determine the presence of significant relationships between the variables, we use a one-sided Student's $t$-test and test for significant differences at the 5% level.

### Derivation of SST indices
To assess the link between the UK droughts and the North Atlantic SST, we constructed two SST indices which describe two SST patterns that have been linked to northern European droughts[20]: SCA and subsequent northward NAC shifts. Both indices were derived by spatially correlating the observed SST anomalies of each month with the specific SST patterns that represent

SCA and northward NAC shifts[20], using the area between 35°N to 65°N and 10°W to 75°W.

For the first index, we correlated the winter season (JFM) SST anomalies for each year with the winter season cold anomaly SST pattern linked to strong freshwater events (Fig. 1j in ref. 20). This index thus represents the time variability of the SCA SST pattern during the JFM season, referred to as SCA-JFM in this study. For the second index, we used the summer season (JJA) SST anomalies for each year correlated with the summer season NAC shift pattern (Fig. 5c in ref. 20). This index—referred to as NAC-JJA—indicates how well the SST pattern in JJA represents the northward NAC shift. The resulting yearly time series for both indices and the associated SST patterns, obtained by regressing the SST back onto the indices, are shown in Fig. 1.

The SCA-JFM pattern (Fig. 1b) resembles the first mode of variability of the North Atlantic SSTs, represented as the North Atlantic tripolar SST pattern positive (negative) phase with cold (warm) SST anomalies in the subpolar region and tropical North Atlantic, with warm (cold) anomalies in between the two. The NAC-JJA pattern (Fig. 1c) is associated with a warm SST anomaly in the region between the subpolar the subtropical gyre, corresponding to the inter-gyre region. It resembles the second mode of variability of the North Atlantic SSTs. The warm anomaly in the inter-gyre region occurs due to a shift in the northward current, resulting in the current (and the warm water it carries) being located anomalously north[52,67,85]. These two patterns were specifically chosen as they have been linked to northern European summer droughts at a lag of 1.5 years to the first index and at zero lag to the second index[20].

## Data availability

Historic gridded SPI for the UK 1862-2015 version 4 is available at UKCEH's Environmental Information Data Centre https://doi.org/10.5285/233090b2-1d14-4eb9-9f9c-3923ea2350ff. The SPI data for years 2016—2022 and observed SSI data are available via the UK Water Resources Portal https://eip.ceh.ac.uk/hydrology/water-resources/. The E.U. Copernicus Marine Service Information's Multi Observation Global Ocean Sea Surface Salinity data is available at https://doi.org/10.48670/moi-00051. ECMWF's ERA5 reanalysis dataset has been downloaded from Copernicus program's Climate Data Store (CDS) https://doi.org/10.24381/cds.f17050d7 and https://doi.org/10.24381/cds.6860a573. The Met Office Hadley Centre Sea Ice and Sea Surface Temperature (HadISST) observed dataset is provided by the UK Met Office Hadley Centre https://www.metoffice.gov.uk/hadobs/hadisst/data/download.html. NAO seasonal index data provided by the Climate Analysis Section, NCAR, Boulder, USA https://climatedataguide.ucar.edu/sites/default/files/2023-07/nao_station_seasonal.txt.

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

## Acknowledgements

This research was funded by the Natural Environment Research Council CANARI project (NE/W004984/1), DIMSUM project (NE/Y005090/1), UKCEH's Net Zero Capacity Building Project, and the Co-Centre for Climate + Biodiversity + Water programme (NE/Y006496/) funded by Science Foundation Ireland (SFI), Northern Ireland's Department of Agriculture,

Environment and Rural Affairs (DAERA) and UK Research and Innovation (UKRI). The authors would like to thank their UKCEH colleague Kate Randall in the Communication Team for producing the illustrations used in Fig. 6.

## Author contributions

All authors contributed to the study design, scientific discussions, and manuscript preparation. A.C. led the data analysis and manuscript writing. M.O., M.T. and B.H. provided datasets. M.O. and B.H. contribute the oceanographic and meteorological perspectives, respectively. M.T., C.S., and J.H. guided the hydrological analysis.

## Competing interests

The authors declare no competing interests.
