## [Peer review file · Communications Earth & Environment]

Oceanic drivers of UK summer droughts

Corresponding Author: Dr Amulya Chevuturi

Version 0:

Decision Letter:

Dear Dr Chevuturi,

Your manuscript titled "Oceanic drivers of UK summer droughts" has now been seen by 2 reviewers, and we include their comments at the end of this message. They find your work of interest, but some important points are raised. We are interested in the possibility of publishing your study in Communications Earth & Environment, but would like to consider your responses to these concerns and assess a revised manuscript before we make a final decision on publication.

We therefore invite you to revise and resubmit your manuscript, along with a point-by-point response that takes into account the points raised. In particular, please discuss in more detail how meteorological and hydrological droughts relate to each other (for example, how often they co-occur), and provide more information on the implications of your work for predictability (quantitative, if possible). Please highlight all changes in the manuscript text file.

Please submit your point-by-point responses as a separate file, distinct from your cover letter where you can add responses to the Editors' comments that you do not want to be made available to the reviewers. Word files are preferred. We recommend that any figures, tables or graphs that are included in the response to reviewers are also included in the main article or Supplementary Information.

Please use the following link to submit your revised manuscript, point-by-point response to the referees' comments (which should be in a separate document to any cover letter), a tracked-changes version of the manuscript (as a PDF file) and the completed checklist:

Link Redacted

We hope to receive your revised paper within six weeks; please let us know if you aren't able to submit it within this time so that we can discuss how best to proceed. If we don't hear from you, and the revision process takes significantly longer, we may close your file. In this event, we will still be happy to reconsider your paper at a later date, as long as nothing similar has been accepted for publication at Communications Earth & Environment or published elsewhere in the meantime.

Please do not hesitate to contact us if you have any questions or would like to discuss these revisions further. We look forward to seeing the revised manuscript and thank you for the opportunity to review your work.

Best regards,

Heike Langenberg, PhD
Chief Editor
Communications Earth & Environment

Editorial Policy: [Policy requirements](https://www.nature.com/documents/nr-editorial-policy-checklist.pdf) (Download the link to your computer as a PDF.)

- Behavioural and social science
- Ecological, evolutionary & environmental sciences
- Life sciences

<https://www.nature.com/documents/nr-reporting-summary.zip>

Furthermore, please align your manuscript with our format requirements, which are summarized on the following checklist: [Communications Earth & Environment formatting checklist](https://www.nature.com/documents/commsj-phys-style-formatting-checklist-article.pdf)

and also in our style and formatting guide [Communications Earth & Environment formatting guide](https://www.nature.com/documents/commsj-phys-style-formatting-guide-accept.pdf) .

*** DATA: Communications Earth & Environment endorses the principles of the Enabling FAIR data project (<http://www.copdess.org/enabling-fair-data-project/>). We ask authors to make the data that support their conclusions available in permanent, publically accessible data repositories. (Please contact the editor if you are unable to make your data available).

All Communications Earth & Environment manuscripts must include a section titled "Data Availability" at the end of the Methods section or main text (if no Methods). More information on this policy, is available at <http://www.nature.com/authors/policies/data/data-availability-statements-data-citations.pdf>.

If a community resource is unavailable, data can be submitted to generalist repositories such as [figshare](https://figshare.com/) or [Dryad Digital Repository](http://datadryad.org/). Please provide a unique identifier for the data (for example a DOI or a permanent URL) in the data availability statement, if possible. If the repository does not provide identifiers, we encourage authors to supply the search terms that will return the data. For data that have been obtained from publically available sources, please provide a URL and the specific data product name in the data availability statement. Data with a DOI should be further cited in the methods reference section.

REVIEWER COMMENTS:

Reviewer #1 (Remarks to the Author):

This is a very well-written paper which provides a valuable and interesting contribution to science around teleconnection influence on UK hydrometeorology. My main comments are around clarity, but I believe most of these can be resolved with ammendments to the text. I have grouped my comments below based on subject.

Page 4, Line 8 (and other locations). I would argue that a pearsons r of 0.3 - 0.4 is not a "strong" connection, certainly in the context of improving forecasting as is discussed later. I would recommend that these assertions are softened. This links into a second comment:

Page 10, 1st and 2nd Para: In my opinion the discussion misses an important step here around combining indicators such

as these with existing modelling or forecasting systems. For instance, any one index may show significant but weak connectivity to hydrometeorological behaviours but can become more powerful when used in combination.

Page 2, Line 4. You mention that the connection between teleconnection and hydrology drought are largely unexplored. I appreciate that this is perhaps in the context of the specific teleconnection you are working with (in which case I might suggest a subtle re-wording), but some of our papers may be of interest where we have demonstrated connections between UK drought measures (including streamflow) and NA teleconnection indices. I am not suggesting these need to be included as citations (although they may add some context) but they may be of interest for the future.

<https://doi.org/10.5194/hess-26-2449-2022>
<https://doi.org/10.1016/j.jhydrol.2024.131831>

Page 4, Line 24. You mention no noticeable auto-correlation in regional SPI3 which I found a bit surprising, particularly in the NW region. It might be beneficial to expand on this a little either in the full text, methods or perhaps SMs as it is a key requirement to much of the subsequent discussion. This has opened up a slight confusion for me... if you find autocorrelations in the SST index, and you find correlations between SST indices and SPI3, would you not expect to find an autocorrelation in SPI3? albeit externally driven. Perhaps this has been handled through the mentions (i.e. partial or residual autocorrelation?) but either way I believe some more explanation around this might help the reader.

Page 5, Line 24-25. I might have misunderstood, but only figure 3c can be explained by groundwater influence? figure 3d appears to be a combination of groundwater and orography? This section might be improved by slightly expanding the discussion / explanation between fig 3c and 3d.

Page 6, Line 11. You mention the statistical relationship may be able to improve forecasting of streamflow drought beyond one-year lead times - but this appears to be based on figure 3 which shows the correlation between SST indices and SPI3 at a one-year lag only? I found this quite confusing as the rest of the paper speaks about a 1.5-year lag. Is this a mistake or was this correlation with streamflow only calculated for a 1-year lag and if so, why not 1.5 year? as this seems to underpin later discussion. Perhaps I missed something but I didn't find a clear explanation for this discrepancy.

Reviewer #2 (Remarks to the Author):

I have gone through the manuscript, and it is technically sound, but I am concerned about the novelty of the work, given the journal is a high-profile journal. The authors have clearly stated, that such teleconnection is already established for meteorological drought and they established for hydrological drought. The real question is whether there was any incident over such a large region when hydrological drought did not follow a meteorological drought. If so, can we have teleconnection as mentioned in the manuscript, if yes, how? This process needs to be explained. If it is routine that hydrological drought in general follows meteorological drought, then it is obvious from previous studies that there will be teleconnections.

My second question is whether this teleconnection adds value to the existing prediction system. If yes, can the authors quantify it? This may add value to the manuscript. At present, it has nice presentation with nicely prepared figures but lacks novelty

Communications Earth & Environment is committed to improving transparency in authorship. As part of our efforts in this direction, we are now requesting that all authors identified as 'corresponding author' create and link their Open Researcher and Contributor Identifier (ORCID) with their account on the Manuscript Tracking System prior to acceptance. ORCID helps the scientific community achieve unambiguous attribution of all scholarly contributions. You can create and link your ORCID from the home page of the Manuscript Tracking System by clicking on 'Modify my Springer Nature account' and following the instructions in the link below. Please also inform all co-authors that they can add their ORCIDs to their accounts and that they must do so prior to acceptance.

Version 1:

Decision Letter:

Dear Dr Chevuturi,

Your manuscript titled "Oceanic drivers of UK summer droughts" has now been seen by our reviewers, whose comments appear below. In light of their advice we are delighted to say that we are happy, in principle, to publish a suitably revised version in Communications Earth & Environment.

We therefore invite you to revise your paper one last time to address the remaining concerns of our reviewers. Specifically, please clarify the remaining points as requested by reviewer 2. At the same time we ask that you edit your manuscript to comply with our format requirements and to maximise the accessibility and therefore the impact of your work.

EDITORIAL REQUESTS:

****Please take care to match our formatting and policy requirements. We will check revised manuscript and return manuscripts that do not comply. Such requests will lead to delays. ****

SUBMISSION INFORMATION:

OPEN ACCESS:

Communications Earth & Environment is a fully open access journal. Articles are made freely accessible on publication. For further information about article processing charges, open access funding, and advice and support from Nature Research, please visit <https://www.nature.com/commsenv/open-access>

Link Redacted

Best regards,

Heike Langenberg, PhD
Chief Editor
Communications Earth & Environment
Communications Sustainability

On Bluesky:
@CommsEarth
@CommsSust

REVIEWERS' COMMENTS:

Reviewer #1 (Remarks to the Author):

I am happy with the justifications and alterations made to the manuscript following my previous set of comments. The updated paper makes an important contribution to the field by elucidating a novel mechanism of hydrological drought development and ultimately for its prediction.

Reviewer #2 (Remarks to the Author):

I appreciate the authors' detailed responses and thank them for their efforts in addressing my comments. I acknowledge their explanation that hydrological drought differs from meteorological drought and agree with this distinction. However, my concern regarding the novelty of the work remains.

Could the authors provide a more in-depth explanation of the reasons behind the differing outcomes of meteorological and hydrological drought? Specifically, why does meteorological drought not always lead to hydrological drought, and vice versa? To what extent does temperature play a role, or could the persistence of medium- to long-term meteorological drought be a key factor in driving severe hydrological drought? A detailed discussion of the underlying physical mechanisms would be helpful.

My second major concern is the novelty of the findings. If teleconnections influence both rainfall and temperature, it seems somewhat expected that they would also influence hydrological drought, given that the latter cannot be forecasted solely based on rainfall. What is the new insight here? Am I overlooking something that makes this result particularly novel?

The authors demonstrate that their teleconnection-based model performs better than operational forecasts and provide a few illustrative examples. However, is there a more comprehensive quantification of its performance? Are there cases where the teleconnection model fails, and if so, what are the reasons? A discussion of its limitations would add clarity.

Finally, I am still searching for a "key" result or conclusion or finding or insight that has a significant impact or demonstrates the transferability of this approach to other regions. Could the authors highlight what makes their findings particularly impactful in this regard?

Open Access This Peer Review File is licensed under a Creative Commons Attribution 4.0 International License, which permits use, sharing, adaptation, distribution and reproduction in any medium or format, as long as you give appropriate credit to the original author(s) and the source, provide a link to the Creative Commons license, and indicate if changes were

made.

Response to Reviewers' comments

Reference: COMMSENV-24-2988-T

Title: Oceanic drivers of UK summer droughts

We thank both the Reviewers and the Editor for taking the time to provide a critical analysis of this manuscript. Specific point-wise replies to the reviewers' comments are provided below, with the corresponding revisions in the manuscript shown as tracked changes as applicable. The changes to the manuscript have improved its clarity and strengthened the scientific arguments presented. We hope that all points raised are addressed satisfactorily below. We look forward to receiving a further decision on this manuscript.

Reviewer#1	
1	This is a very well-written paper which provides a valuable and interesting contribution to science around teleconnection influence on UK hydrometeorology. My main comments are around clarity, but I believe most of these can be resolved with ammendments to the text. I have grouped my comments below based on subject.
	We thank the reviewer for the positive review and appreciate the constructive suggestions. All of these have improved the results and discussion of our work.
2	Page 4, Line 8 (and other locations). I would argue that a pearsons r of 0.3 - 0.4 is not a "strong" connection, certainly in the context of improving forecasting as is discussed later. I would recommend that these assertions are softened.
	We agree with the reviewer that these correlation coefficient values do not indicate "strong connections". Accordingly, we have softened our assertions in the revised manuscript and now refer to them as "significant correlations" Section 2 Paragraph 2. However, while these correlation values may appear low, it is important to note that area-averaging SPI over large regions tends to weaken the signal. This effect is particularly relevant because SST patterns influence UK rainfall, yet the spatial distribution of rainfall varies slightly each year depending on the exact positioning of the SST front between the warm North Atlantic current and the subpolar cold anomaly (Oltmanns et al., 2024). By averaging over many years which all have slightly shifted SST fronts relative to each other, the correlation goes down as the number of degrees of freedom goes up. Oltmanns et al. (2024) suggest that the correlations increase substantially (up to 0.8 or 0.9) when restricting the analysis to only a subset of years which all have similar SST front locations. To provide a more comprehensive analysis, we have examined spatial correlation patterns and specific case studies, as these SST patterns strongly influence drought

	indicators. To clarify this, we have added additional details in the revised manuscript's Section 2 Paragraph 2. In the context of using these indicators for forecasting, while the correlation coefficients may not appear very strong, they can outperform current operational hydrological forecasting systems for northwest UK at such long lead times, which has no skill beyond the southeast UK. This highlights their potential to add value to the forecasting system. Further, these indicators are useful for predicting extreme drought events, as demonstrated by our work, enabling the development of regime-dependent forecasting systems with improved accuracy for low flows. Based on your current and subsequent comments, we have incorporated this point into the revised manuscript Section 4 Paragraph 5. To further illustrate the significance of these indicators and the contribution of our work, we refer to our response to Reviewer#2 Comment#2. There, we present examples of operational seasonal forecasts for the 2018 and 2022 droughts, which failed to capture the severity of these events over the UK, even with a three-month lead time. In contrast, our case study analysis in Section 3 demonstrates that North Atlantic freshwater events provided clear early signals of these droughts well in advance. This evidence suggests that SST indicators derived from freshwater events in the North Atlantic offer predictability for improving UK hydrological forecasts, particularly for drought prediction.  • Oltmanns et al. (2024) European summer weather linked to North Atlantic freshwater anomalies in preceding years, Weather and Climate Dynamics, 5(1) 109-132, https://doi.org/10.5194/wcd-5-109-2024
3	This links into a second comment: Page 10, 1st and 2nd Para: In my opinion the discussion misses an important step here around combining indicators such as these with existing modelling or forecasting systems. For instance, any one index may show significant but weak connectivity to hydrometeorological behaviours but can become more powerful when used in combination.
	We thank the reviewer for this valuable suggestion. As mentioned in our previous reply, our analysis demonstrates that these indicators are particularly effective for low-flow regimes, especially in northwest UK, where current hydrological forecasting systems exhibit lower skill—and vice-versa. Thus, following the reviewer's suggestion, combining these indicators with existing hydrological forecasting systems would be a promising approach. Additionally, different links within the teleconnection system can serve as sequential predictors at varying lead times, enhancing forecasting accuracy, as the prediction period approaches, and lead time reduces. Similar method was applied to drought forecasting over Australia using attention models (Deo et al., 2017; Dikshit et al., 2022). This type of approach would be promising in improving upon any predictions based on a single teleconnection system, but is beyond the scope of this study, and would constitute an avenue for future work.

	To enhance the discussion related to drought forecasting systems, we have incorporated a more in-depth discussion in the last paragraph of Section 4 in the revised manuscript.  • Deo et al. (2017) Drought forecasting in eastern Australia using multivariate adaptive regression spline, least square support vector machine and M5Tree model, Atmospheric Research, 184, 149-175, https://doi.org/10.1016/j.atmosres.2016.10.004. • Dikshit et al. (2022) Solving transparency in drought forecasting using attention models, Science of The Total Environment, 837, 155856, https://doi.org/10.1016/j.scitotenv.2022.155856.
4	Page 2, Line 4. You mention that the connection between teleconnection and hydrology drought are largely unexplored. I appreciate that this is perhaps in the context of the specific teleconnection you are working with (in which case I might suggest a subtle re-wording), but some of our papers may be of interest where we have demonstrated connections between UK drought measures (including streamflow) and NA teleconnection indices. I am not suggesting these need to be included as citations (although they may add some context) but they may be of interest for the future. https://doi.org/10.5194/hess-26-2449-2022 https://doi.org/10.1016/j.jhydrol.2024.131831
	We do mean to only refer to specific teleconnections from the North Atlantic SSTs (associated with the freshwater events) to UK hydrology here. We have re-worded the sentence accordingly. We thank the reviewer for the references provided, we have added these to the Section 1 Paragraph 2, describing the influence of NAO on UK droughts to provide further context.
5	Page 4, Line 24. You mention no noticeable auto-correlation in regional SPI3 which I found a bit surprising, particularly in the NW region. It might be beneficial to expand on this a little either in the full text, methods or perhaps SMs as it is a key requirement to much of the subsequent discussion. This has opened up a slight confusion for me... if you find autocorrelations in the SST index, and you find correlations between SST indices and SPI3, would you not expect to find an autocorrelation in SPI3? albeit externally driven. Perhaps this has been handled through the mentions (i.e. partial or residual autocorrelation?) but eitherway I believe some more explanation around this might help the reader.
	We agree that there is a correlation between SST indices and SPI3, and that SST indicators themselves exhibit auto-correlation. However, this may not necessarily cause SPI3 to show auto-correlation. Please see Figure 8d in Oltmanns et al. (2024) for differences in autocorrelations of SST and precipitation minus evaporation. Further, it is worth bearing in mind that the actual precipitation amount that

	reaches the ground is highly influenced/modified by the temporary characteristics of the local area over which it is formed, and this will affect even a regional 3-month aggregation. Hence, SPI3 exhibits high variability, and its variance is influenced by many forcings including more short-term and/or local wind direction (e.g., from SST source area, topographical windward/leeward effects) and land temperatures and SSTs in other areas. Therefore, while our particular SST indicators affect SPI3, they do not account for its full variance and thus will not necessarily induce auto-correlation in SPI3. As per the reviewer's suggestion, we have provided a more detailed explanation in the revised manuscript Section 2 Paragraph 3.  • Oltmanns et al. (2024) European summer weather linked to North Atlantic freshwater anomalies in preceding years, Weather and Climate Dynamics, 5(1) 109-132, https://doi.org/10.5194/wcd-5-109-2024
6	Page 5, Line 24-25. I might have misunderstood, but only figure 3c can be explained by groundwater influence? figure 3d appears to be a combination of groundwater and orography? This section might be improved by slightly expanding the discussion / explanation between fig 3c and 3d.
	The reviewer raises a valid point that Figure 3d reflects influences beyond groundwater alone. We have now amended by also discussing the effects of orography and evaporative demand in the revised manuscript in Section 2 Paragraph 7 (penultimate paragraph of Section 2).
7	Page 6, Line 11. You mention the statistical relationship may be able to improve forecasting of streamflow drought beyond one-year lead times - but this appears to be based on figure 3 which shows the correlation between SST indices and SPI3 at a one-year lag only? I found this quite confusing as the rest of the paper speaks about a 1.5-year lag. Is this a mistake or was this correlation with streamflow only calculated for a 1-year lag and if so, why not 1.5 year? as this seems to underpin later discussion. Perhaps I missed something but I didn't find a clear explanation for this discrepancy.
	We apologise for any confusion. The 1.5-year lag refers to the period from the SCA-JFM (Year 0) to the SSI3/SPI3 JJA (Year +1), indicating that the teleconnection spans from the winter of the preceding year to the summer of the following year. To make this clear, we have added a line in Section 2 Paragraph 5.

Reviewer#2	
1	I have gone through the manuscript, and it is technically sound, but I am concerned about the novelty of the work, given the journal is a high-profile journal. The authors have clearly stated, that such teleconnection is already established for meteorological

drought and they established for hydrological drought. The real question is whether there was any incident over such a large region when hydrological drought did not follow a meteorological drought. If so, can we have teleconnection as mentioned in the manuscript, if yes, how? This process needs to be explained. If it is routine that hydrological drought in general follows meteorological drought, then it is obvious from previous studies that there will be teleconnections.

We thank the reviewer for their valuable suggestions and for highlighting the technical authenticity of our work.

While we acknowledge the reviewer's point that there is an approximate relationship between the major historical meteorological and hydrological droughts, it is incorrect to assume that analysing meteorological droughts alone provides sufficient insight into hydrological droughts without explicitly evaluating the hydrological impacts. As demonstrated by Laaha et al. (2017) for Europe, hydrological responses can differ significantly over different regions even with similar meteorological drought conditions. Studies over UK comparing hydrological (Figure R1), and meteorological (Figure R2) droughts ranked by severity reveals differences in the respective droughts ranking periods. For example, the 1933–1935 meteorological drought ranked first in the SE region but was not the highest-ranked drought in a hydrological context. On the other hand, the drought of the early 1960s was a severe hydrological drought but did not rank highly as meteorological droughts. Therefore, studying teleconnections solely to UK rainfall is insufficient, as hydrological droughts may not follow the same patterns. It is thus essential for our study to examine teleconnection links in the context of UK hydrology.

The concept of drought propagation (i.e., from meteorological droughts to hydrological droughts) means that different catchments (and regions) will display very different hydrological responses even to the same meteorological forcings, as catchment properties (e.g., land surface properties, storage in soils, and groundwater) influence hydrological drought properties (their frequency, duration, severity and even seasonality) and cause them to diverge significantly from the meteorological drivers (Loon et al., 2015). The role of catchment properties in influencing the propagation from meteorological to hydrological drought is covered extensively in the international literature, and quantified in some detail for the UK by Barker et al. (2016). Even our study also reveals distinct differences between meteorological and hydrological responses to North Atlantic SST indicators, as shown in Figures 3a and 3c of our manuscript. The southeast UK exhibits a slow hydrological response, suggesting that these indicators are effective for developing skilful streamflow forecasting systems only in regions outside the southeast. However, for precipitation, these indicators influence the entire UK.

Figure R1: Figure 5 from Barker et al. (2019) showing the top 10 extracted events from SSI-12, using a threshold of -1.5 for each drought event characteristic. Catchments are arranged roughly from north to south on the y axis, with each row representing a catchment and hydro-climatic regions marked for clarity. Bars represent the top 10 events and are coloured according to the event rank; darker shades represent higher-ranking (i.e. more severe) events.

Hydrological droughts in the UK are important indicators of water resource availability and have direct impact on the environment and society. Many studies have previously argued the importance of hydrological droughts impacts (e.g., Cloke et al., 2011; Pozzi et al., 2013; Hao et al., 2018). Further, we believe Reviewer#1 acknowledges this point, as evidenced by the literature cited (Rust et al., 2022; 2024) in their Comment#4, which examines the teleconnections between the NAO and UK hydrology (streamflow and groundwater). These studies specifically focus on the impacts on water resources rather than only on meteorological factors. Identifying long lead-time teleconnections for hydrological droughts is very useful, as it represents the first step in developing predictive systems that provide early warnings, allowing the water managers time to mobilise resources and implement anticipatory measures to mitigate severe impacts of

droughts. With climate change projected to increase the frequency and intensity of extreme hydrological droughts in the coming years, this need becomes even more pressing. Relying on meteorological drought indicators may not be sufficient and could undermine confidence in forecasting systems if the impacts on water resources are not adequately considered.

The above points provide a response to the reviewer's specific concern over there not being sufficient novelty in moving from meteorological to hydrological drought. We also want to highlight that our work has sufficient novelty in general, over and above this. While Oltmanns et al. (2024) demonstrate statistical links between North Atlantic indicators and meteorological droughts, but they do not evaluate the impacts on hydrological droughts. They also did not delve into the details of each link of the teleconnection pathway. Additionally, other studies cited in our work have examined specific segments of the teleconnection pathway; for example, Kingston et al. (2015) explored the relationship between the NAO and European meteorological droughts. However, no study to date has fully traced the entire teleconnection pathway, from North Atlantic salinity changes to UK hydrology. The novelty of our paper lies in not only demonstrating the statistical links to UK hydrology but also connecting each segment of the teleconnection pathway. We trace the connections from freshwater incursion into the North Atlantic to SST changes, through atmospheric patterns (NAO and Jet Stream), and ultimately to meteorological and hydrological droughts. By combining these statistical links with robust evidence from a comprehensive literature review of the various components of the teleconnection, along with demonstration of drought case studies, our work offers a novel and unprecedented analysis that has not been presented before.

For the reasons outlined above, our paper is not only novel but has the potential to become a seminal work in exploring the teleconnection between North Atlantic freshwater events and real-world impacts on water resource availability due to hydrological droughts. We hope that our study sparks further in-depth research into this teleconnection pathway as a whole, rather than focusing on individual segments. Such future studies would ultimately contribute to the development of a prediction system for hydrological droughts, grounded not only in statistical links but also in a robust process-based understanding of the underlying mechanisms. This combined approach will enhance confidence in the prediction system, making it more reliable and likely to be adopted by water managers, where its implications could underpin significant financial decision-making and ultimately improved societal outcomes. Finally, we reaffirm the point made in our paper that the paper's novelty and potential impact extends beyond the UK – the innovative full-chain analysis of the teleconnection pathways could readily be applied elsewhere.

We agree with the reviewer's point that this point may have not been clearly emphasised in the previous version of our manuscript. To clearly highlight the novelty of our work and emphasize the importance of tracing the teleconnection pathway from North Atlantic freshwater events to hydrological droughts in greater detail, we have included an in-depth discussion in Section 1, Paragraphs 5 and 6 of our revised manuscript.

	 • Barker et al. (2016) From meteorological to hydrological drought using standardised indicators, Hydrology and Earth System Sciences, 20, 2483–2505, https://doi.org/10.5194/hess-20-2483-2016. • Barker et al. (2019) Historic hydrological droughts 1891–2015: systematic characterisation for a diverse set of catchments across the UK, Hydrology and Earth System Sciences, 23, 4583–4602, https://doi.org/10.5194/hess-23-4583-2019. • Cloke et al. (2011) Large-scale hydrology: Advances in understanding processes, dynamics and models from beyond river basin to global scale. Hydrological Processes, 25, 991–995, https://doi.org/10.1002/hyp.8059. • Hao et al. (2018). Seasonal drought prediction: Advances, challenges, and future prospects. Reviews of Geophysics, 56, 108–141. https://doi.org/10.1002/2016RG000549. • Kingston et al. (2015) European-Scale Drought: Understanding Connections between Atmospheric Circulation and Meteorological Drought Indices, Journal of Climate, 28, 505–516, https://doi.org/10.1175/JCLI-D-14-00001.1. • Laaha et al. (2017) The European 2015 drought from a hydrological perspective, Hydrology and Earth System Sciences, 21, 3001–3024, https://doi.org/10.5194/hess-21-3001-2017. • Loon et al. (2015) Hydrological drought explained, WIREs Water, 2, 359–392, https://doi.org/10.1002/wat2.1085. • Oltmanns et al. (2024) European summer weather linked to North Atlantic freshwater anomalies in preceding years, Weather and Climate Dynamics, 5(1) 109–132, https://doi.org/10.5194/wcd-5-109-2024 • Pozzi et al. (2013) Towards global drought early warning capability: Expanding international cooperation for the development of a framework for global drought monitoring and forecasting. Bulletin of the American Meteorological Society, 94, 776–785, https://doi.org/10.1175/BAMS-D-11-00176.1. • Tanguy et al. (2021) Regional Differences in Spatiotemporal Drought Characteristics in Great Britain, Frontiers in Environmental Science, 9, https://doi.org/10.3389/fenvs.2021.639649. • Rust et al. (2022) The importance of non-stationary multiannual periodicities in the North Atlantic Oscillation index for forecasting water resource drought, Hydrology and Earth System Sciences, 26, 2449–2467, https://doi.org/10.5194/hess-26-2449-2022. • Rust et al. (2024) Long-range hydrological drought forecasting using multi-year cycles in the North Atlantic Oscillation, Journal of Hydrology, 641, 131831, https://doi.org/10.1016/j.jhydrol.2024.131831.
2	My second question is whether this teleconnection adds value to the existing prediction system. If yes, can the authors quantify it? This may add value to the manuscript. At present, it has nice presentation with nicely prepared figures but lacks novelty
	We thank the reviewer for their positive feedback on the presentation of our figures and manuscript. Regarding the reviewer’s concern about the novelty of our work, we

refer to our detailed response to the previous comment. We believe we have clearly demonstrated the novelty of our research while also highlighting our study's real-world significance, particularly in the context of long lead-time drought predictions for water scarcity and resource management.

This teleconnection enhances the current operational hydrological prediction systems, particularly for regions outside southeast UK, and is especially effective for low-flow regimes. For a more detailed response, please refer to our reply to Reviewer#1, Comment#3. To further emphasize this point, we have expanded the discussion in the final paragraph of Section 4 in the revised manuscript.

In our study, the predictability from North Atlantic SST indicators has been quantified using correlation coefficients, providing a foundation for future research aimed at developing predictive systems. As stated in the original version of manuscript, "*to identify a clear causal chain of events, SST-forced simulations can be performed with prescribed observed SST and sea ice.*" However, to actually perform such experiments is beyond the current scope of our study. Additionally, to quantitatively assess the impact of a prediction system using these indicators, such a system would first need to be developed, which we also consider beyond the scope of this work.

Currently, freshwater feedback mechanisms are generally underrepresented in models (e.g., Menary et al., 2015; Sgubin et al., 2017; Mecking et al., 2017; Wu et al., 2018) and need to be integrated into more advanced modelling frameworks. Through our work, we aim to raise awareness within the modelling community about the importance of incorporating these freshwater signals, which are presently overlooked. We have reinforced this point by including this in the Paragraph 6 of Section 1 of the revised manuscript.

To illustrate how this teleconnection is not accounted for in current prediction systems, we examine past forecasts for the extreme UK drought events of 2018 and 2022. The meteorological seasonal forecasting systems from ECMWF and the UK Met Office did not indicate a high probability of lower-tercile precipitation even at 3- to 4-month lead times (Figure R3). The operational UK Hydrological Outlooks forecasted some below-normal summer river flows for 2018 and 2022 (Figure R4) in May but did not capture the full severity of the droughts, even at such short lead times. However, as demonstrated in Section 3 of our manuscript, the teleconnection pathway identified in our study could have predicted the droughts of 2018 and 2022 well in advance.

Figure R3: (a) ECMWF seasonal forecast using SEAS5 system from at 4-month lead time for with base time (initialisation) of Feb 2018 to forecast summer (JJA) 2018 (https://charts.ecmwf.int/products/seasonal_system5_standard_rain?area=EURO&base_time=201802010000&stats=tsum&valid_time=201806040000). (b) is same as (a) but with base time (initialisation) of Feb 2022 to forecast summer (JJA) 2022 (https://charts.ecmwf.int/products/seasonal_system5_standard_rain?area=EURO&base_time=202202010000&stats=tsum&valid_time=202206040000). (c) UK Met Office seasonal forecast using GloSea6 system from at 3-month lead time for with base time (initialisation) of March 2018 to forecast summer (JJA) 2018 (https://climate.copernicus.eu/charts/packages/c3s_seasonal/products/c3s_seasonal_spatial_eqr_rain_3m?area=area01&base_time=201803010000&type=tsum&valid_time=201806010000). (d) is same as (c) but with base time (initialisation) of March 2022 to forecast summer (JJA) 2022 (https://climate.copernicus.eu/charts/packages/c3s_seasonal/products/c3s_seasonal_spatial_eqr_rain_3m?area=area01&base_time=202203010000&type=tsum&valid_time=202206010000).

(a) 2018

(b) 2022

Figure R4: UK Hydrological Outlook for the base time from (a) May 2018 (https://nora.nerc.ac.uk/id/eprint/522903/1/2018_05_HO_Summary.pdf) and (b) May 2022 (https://nora.nerc.ac.uk/id/eprint/532874/1/2022_05_HO_Summary.pdf) to discuss the forecasting for 1-3 upcoming month respectively.

- Mecking et al. (2017) The effect of model bias on Atlantic freshwater transport and implications for AMOC bi-stability, *Tellus A*, 69, 1299910, <https://doi.org/10.1080/16000870.2017.1299910>.
- Menary et al. (2015) Exploring the impact of CMIP5 model biases on the simulation of North Atlantic decadal variability, *Geophysical Research Letters*, 42, 5926–5934, <https://doi.org/10.1002/2015GL064360>.
- Sgubin et al. (2017) Abrupt cooling over the North Atlantic in modern climate models, *Nature Communications*, 8, 14375, <https://doi.org/10.1038/ncomms14375>.
- Wu et al. (2018) North Atlantic climate model bias influence on multiyear predictability, *Earth and Planetary Science Letters*, 481, 171–176, <https://doi.org/10.1016/j.epsl.2017.10.012>.

Response to Reviewers' comments

Reference: COMMSENV-24-2988A

Title: Oceanic drivers of UK summer droughts

We thank both reviewers for taking the time to provide feedback on the manuscript and our previous replies. Specific point-wise replies to the latest round of reviewers' comments are provided below, with the corresponding revisions in the manuscript shown as tracked changes as applicable. We hope that all points raised are addressed satisfactorily below. We look forward to receiving a further decision on this manuscript.

Reviewer#1	
1	I am happy with the justifications and alterations made to the manuscript following my previous set of comments. The updated paper makes an important contribution to the field by elucidating a novel mechanism of hydrological drought development and ultimately for its prediction.
	We thank the reviewer for the positive feedback. We really appreciate all their efforts in improving this manuscript.

Reviewer#2	
1	I appreciate the authors' detailed responses and thank them for their efforts in addressing my comments. I acknowledge their explanation that hydrological drought differs from meteorological drought and agree with this distinction. However, my concern regarding the novelty of the work remains.
	We thank the reviewer for their feedback. Regarding the comment on the novelty of our work, please refer to our response to Reviewer#2 Comment#3 and Reviewer#2 Comment#5 for a detailed explanation.
2	Could the authors provide a more in-depth explanation of the reasons behind the differing outcomes of meteorological and hydrological drought? Specifically, why does meteorological drought not always lead to hydrological drought, and vice versa? To what extent does temperature play a role, or could the persistence of medium- to long-term meteorological drought be a key factor in driving severe hydrological drought? A detailed discussion of the underlying physical mechanisms would be helpful.
	Before addressing the reviewer's query in detail, we would like to refer to our comprehensive response to Reviewer#2 Comment#1 in the previous review cycle, where we discussed the differences between meteorological and hydrological droughts.

While meteorological droughts contribute to hydrological droughts in the short- as well as long-term, they are not the only influencing factor (Sutanto et al., 2024) and research has shown that globally only some meteorological droughts propagate into hydrological droughts (Kumar et al., 2025). One reason meteorological droughts may not necessarily lead to hydrological droughts is due to catchment characteristics like storage properties and response times (e.g., Fleig et al., 2011). In regions with significant groundwater influence, such as the southeast of the UK, groundwater contributions can buffer the impacts of reduced precipitation, preventing the development of hydrological droughts. These groundwater-dominated catchments do not respond directly to meteorological conditions, resulting in a decoupling between precipitation and river flows (e.g., Lavers et al., 2010). In the UK, only prolonged multi-year meteorological droughts transition into hydrological droughts within groundwater-dominated catchments (Barker et al., 2016). Consequently, as shown in our study while the North Atlantic freshwater events strongly influence precipitation in the southeast UK, they show little to no significant correlation with river flows. This distinct hydrological response highlights the need to study hydrological droughts in their own right, rather than treating them merely as a direct consequence of meteorological droughts.

Temperature influences hydrological drought through its effects on snow water accumulation, the timing of snowmelt, and evaporative demand (Brunner et al., 2021; Ahmadalipour et al., 2017). In this context, temperature primarily affects the proportion of rainfall that ultimately contributes to river flow (Rahmani & Fattahi, 2021). However, other factors, such as percolation into groundwater, also play a role (van Lanen et al., 2004). Additionally, evapotranspiration is not determined by temperature alone; variables like wind speed and cloudiness are also important (Teuling et al., 2013).

Thus, the propagation of drought across different types: meteorological, soil moisture (or agricultural), hydrological, and socio-economic, is inherently complex. While both temperature, evaporation and precipitation play important roles in the development of hydrological droughts, their influence varies depending on catchment characteristics modulating the influence of climatic drivers (Kumar et al., 2025). The processes that govern the transition from meteorological to hydrological drought can be broadly categorized into *lag* and *attenuation* (which are mainly controlled by catchment properties), and *pooling* and *lengthening* (which are influenced by both catchment and climate factors). Meteorological to hydrological droughts propagation processes include: multiple meteorological droughts may combine (pooling) into a single prolonged hydrological drought; their intensity is reduced (attenuation) by catchment storage; onset is delayed (lag) across different drought types; and overall duration increases (lengthening) from meteorological to hydrological drought. These mechanisms are discussed in detail by Van Loon et al. (2015) but are beyond the scope of our study.

Hence, despite the strong relationship between rainfall and river flows, this relationship is confounded by other factors and correlation between meteorological and hydrological droughts can often be significantly diminished, as various catchment

processes and other local climate conditions influence drought propagation (Hannaford et al., 2011; Kumar et al., 2025). In consequence, when large-scale climate drivers such as teleconnections are considered, meteorological indicators alone are insufficient proxies for hydrological drought, particularly in certain types of catchments, as demonstrated in not only our study but many others previously (e.g., Laizé & Hannah, 2010; Fleig et al., 2011; Rust et al., 2022).

To emphasise the above discussed points in our study, we have cited additional literature and added more details in Section 1 Paragraph 5 of the revised version of our manuscript.

References:

- Ahmadalipour, A., Moradkhani, H., & Demirel, M. C. (2017). A comparative assessment of projected meteorological and hydrological droughts: elucidating the role of temperature. *Journal of Hydrology*, 553, 785-797.
- Barker, L. J., Hannaford, J., Chiveron, A. & Svensson, C. (2016) From meteorological to hydrological drought using standardised indicators. *Hydrology and Earth System Sciences*, 20, 2483–2505.
- Brunner, M. I., Swain, D. L., Gilleland, E., & Wood, A. W. (2021). Increasing importance of temperature as a contributor to the spatial extent of streamflow drought. *Environmental Research Letters*, 16(2), 024038.
- Fleig, A. K., Tallaksen, L. M., Hisdal, H., & Hannah, D. M. (2011). Regional hydrological drought in north-western Europe: linking a new Regional Drought Area Index with weather types. *Hydrological Processes*, 25(7), 1163-1179.
- Hannaford, J., Lloyd-Hughes, B., Keef, C., Parry, S., & Prudhomme, C. (2011). Examining the large-scale spatial coherence of European drought using regional indicators of precipitation and streamflow deficit. *Hydrological Processes*, 25(7), 1146-1162.
- Kumar, A., Gosling, S. N., Johnson, M. F., Jones, M. D., Nkwasa, A., Koutroulis, A., Müller Schmied, H., Li, H.-Y., Kim, H., Hanasaki, N., Kumar, R., Thiery, W. & Pokhrel, Y. (2025). Cascading droughts: Exploring global propagation of meteorological to hydrological droughts (1971–2001). *Science of The Total Environment*, 979, 179486.
- Laizé, C. L., & Hannah, D. M. (2010). Modification of climate–river flow associations by basin properties. *Journal of Hydrology*, 389(1-2), 186-204.
- Lavers, D., Prudhomme, C., & Hannah, D. M. (2010). Large-scale climate, precipitation and British river flows: Identifying hydroclimatological connections and dynamics. *Journal of Hydrology*, 395(3-4), 242-255.
- Rahmani, F., & Fattahi, M. H. (2021). A multifractal cross-correlation investigation into sensitivity and dependence of meteorological and hydrological droughts on precipitation and temperature. *Natural Hazards*, 109(3), 2197-2219.
- Rust, W., Bloomfield, J. P., Cuthbert, M., Corstanje, R., & Holman, I. (2022). The importance of non-stationary multiannual periodicities in the North Atlantic

	Oscillation index for forecasting water resource drought. Hydrology and Earth System Sciences, 26(9), 2449-2467. Sutanto, S. J., Syaehuddin, W. A., & de Graaf, I. (2024). Hydrological drought forecasts using precipitation data depend on catchment properties and human activities. Communications Earth & Environment, 5(1), 118. Teuling, A. J., Van Loon, A. F., Seneviratne, S. I., Lehner, I., Aubinet, M., Heinesch, B., Bernhofer, C., Grünwald, T., Prasse, H., & Spank, U. (2013). Evapotranspiration amplifies European summer drought. Geophysical Research Letters, 40(10), 2071-2075. van Lanen, H. A. J., Fendeková, M., Kupczyk, E., Kasprzyk, A., & Pokojski, W. (2004). Flow generating processes. In L. M. Tallaksen, & H. A. J. van Lanen (Eds.), Hydrological Drought. Processes and estimation methods for streamflow and groundwater (pp. 53-96). (Developments in Water Science; No. 48). Van Loon, A. F. (2015). Hydrological drought explained. Wiley Interdisciplinary Reviews: Water, 2(4), 359-392.
3	My second major concern is the novelty of the findings. If teleconnections influence both rainfall and temperature, it seems somewhat expected that they would also influence hydrological drought, given that the latter cannot be forecasted solely based on rainfall. What is the new insight here? Am I overlooking something that makes this result particularly novel?
	We would like to draw your attention to our detailed response to Reviewer#2 Comment#1 from the previous review cycle, as well as our reply to Reviewer#2 Comment#5 in the current document, where we outline the novelty of our work and highlight the key results respectively. In our response to Reviewer#2 Comment#2, we clearly explained that tracing teleconnections only to meteorological droughts is insufficient for understanding hydrological droughts, which directly affect water resource availability. Our study demonstrates this: while the teleconnection influences precipitation in the Southeast UK, it does not have a corresponding impact on river flows. If we had focused solely on rainfall, the impact on the Southeast region would have been misrepresented, leading to the model developed using this teleconnection to underperform in this area and, ultimately, undermining confidence among water managers. Although the propagation of drought from meteorology to hydrology is strongly influenced by precipitation and temperature, it also depends on other local climate factors and the catchment dynamics. Therefore, relying solely on precipitation and temperature as indicators for river flow, and by extension, hydrological drought, would be insufficient. To further highlight the novelty of our work, we have added the discussed points into the Introduction Section 1 Paragraph 5 of the revised manuscript.

4	The authors demonstrate that their teleconnection-based model performs better than operational forecasts and provide a few illustrative examples. However, is there a more comprehensive quantification of its performance? Are there cases where the teleconnection model fails, and if so, what are the reasons? A discussion of its limitations would add clarity.
	In this work while we have not yet developed a fully operational model, our study lays the groundwork by introducing and establishing the key teleconnections driver for UK summer droughts. Comprehensive evaluation of model performance lies beyond the current scope, but our findings offer a new approach to understanding, and ultimately predicting, these events, providing a solid foundation for future work on developing a predictive model. Our findings quantify the strength of the teleconnections through linear correlations, which are statistically significant and also provide a measure of the explained variance. A key limitation, however, is the limited number of extreme UK summer drought events in the historical observed record, restricting our ability to fully specify the involved sensitivities. Additionally, while our correlations explain part of the drought variability, they do not capture it entirely. We anticipate that these relationships can be refined through (1) the development of a multiple-linear regression model, also considering other drivers of droughts, (2) additional sensitivity analyses to detect potential nonlinearities, and (3) the identification and ultimately removal of biases in existing forecasting methods, thus improving the accuracy of UK summer drought forecasts overall. Our study provides the groundwork and motivation for further analyses, but unfortunately, we cannot (yet) provide the complete forecasting model. While these limitations were previously mentioned in our manuscript, we have expanded on them further in Section 4 Paragraph 6 of the revised version.
5	Finally, I am still searching for a "key" result or conclusion or finding or insight that has a significant impacts or demonstrates the transferability of this approach to other regions. Could the authors highlight what makes their findings particularly impactful in this regard?
	Significant impacts from our findings have already been described clearly in our manuscript's Section 4 Discussion & Conclusion Paragraph 5. Our key finding reveals the full chain of teleconnections linking North Atlantic freshwater events to UK hydrology, offering significant potential to enhance long lead-time predictions of UK summer droughts. Our work has unique, significant and novel implications which are summarised below:  1. Our work has identified long lead-time teleconnections that can reliably predict extreme summer droughts in the UK from a hydrological perspective. A point to note is that the relationship between North Atlantic drivers and river flows cannot be assumed to mirror that with precipitation, especially in regions like the Southeast UK, where the teleconnection link holds for precipitation but not for river flows. This finding has direct implications for

water resource planning, as accurate early forecasts are essential for water managers to prepare and mitigate the impacts of droughts well in advance.

2. Our study is the first to trace the full chain of events, from freshwater input into the North Atlantic to UK summer droughts, offering a novel and comprehensive description of this teleconnection. Unlike previous studies that focus on isolated links, our work establishes a causal and dynamic connection across the entire pathway. This deeper understanding can underpin the development of more reliable long lead-time prediction systems, which will increase the confidence in the forecasts they produce. With climate change accelerating ice melt, freshwater events in the North Atlantic are likely to become more frequent and more intense, making it increasingly important to understand these teleconnections in the context of drought risk.
3. Our study is the first to deliver continuous and easily obtainable SST indicators of UK droughts in any given year and is thus more accessible to wider audiences than previous work focusing on individual links. The SST indices can be used by the scientific community to expand and refine the forecasting approach for potential end users for different use cases (e.g., agriculture, environmental management, water resource management, insurance) and they can also easily be translated to global climate model analyses for future model improvements.

We have further emphasized the two points discussed above in the Introduction and the Discussion & Conclusion sections of the revised manuscript.

Finally, this work has broader relevance beyond the UK, with the potential for this approach to be transferred to other regions, as hydrological forecasting systems are becoming increasingly essential, particularly in the context of future climate change, which is expected to intensify drought risk (Sutanto et al., 2020). Initial results from our ongoing follow-up study indicate that these teleconnections also have a significant impact on European hydrological droughts, including both river flows and groundwater levels. We also find that the impact and predictability arising from freshwater events in the North Atlantic and associated SST anomalies is sensitive to the exact location, strength and extent of the freshwater (and SST) anomalies and their subsequent ocean-atmosphere feedbacks. Thus, the UK is by no means an exception. Current climate models substantially underestimate the predictability in the large-scale North Atlantic region, not only in the UK (e.g., Scaife & Smith, 2018; Shaw et al., 2024; Weisheimer et al., 2024). Our study shows a new forecasting potential that also applies to other European regions. We have included this point in the Section 4 Paragraph 5 of the revised manuscript.

References:

Scaife, A. A., & Smith, D. (2018). A signal-to-noise paradox in climate science. *NPJ Climate and Atmospheric Science*, 1(1), 28.

Shaw, T. A., Arias, P. A., Collins, M., Coumou, D., Diedhiou, A., Garfinkel, C. I., Jain, S., Roxy, M., K., Kretschmer, M., Leung, L., Narsey, S., Martius, O., Seager, R., Shepherd, T.

G., Sörensson, A. A., Stephenson, T., Taylor, M. & Wang, L. (2024). Regional climate change: consensus, discrepancies, and ways forward. *Frontiers in Climate*, 6, 1391634.

Sutanto, S. J., Wetterhall, F., & Van Lanen, H. A. (2020). Hydrological drought forecasts outperform meteorological drought forecasts. *Environmental Research Letters*, 15(8), 084010.

Weisheimer, A., Baker, L. H., Bröcker, J., Garfinkel, C. I., Hardiman, S. C., Hodson, D. L., Palmer, T. N., Robson, J. I., Scaife, A. A., Screen, J. A., Shepherd, T. G., Smith, D. M. & Sutton, R. T. (2024). The signal-to-noise paradox in climate forecasts: revisiting our understanding and identifying future priorities. *Bulletin of the American Meteorological Society*, 105(3), E651-E659.